# PUREE: accurate pan-cancer tumor purity estimation from gene expression data

Egor Revkov [1,2], Tanmay Kulshrestha[1], Ken Wing-Kin Sung[1,2] & Anders Jacobsen Skanderup [1,2,3 ✉]

Tumors are complex masses composed of malignant and non-malignant cells. Variation in tumor purity (proportion of cancer cells in a sample) can both confound integrative analysis and enable studies of tumor heterogeneity. Here we developed PUREE, which uses a weakly supervised learning approach to infer tumor purity from a tumor gene expression profile. PUREE was trained on gene expression data and genomic consensus purity estimates from 7864 solid tumor samples. PUREE predicted purity with high accuracy across distinct solid tumor types and generalized to tumor samples from unseen tumor types and cohorts. Gene features of PUREE were further validated using single-cell RNA-seq data from distinct tumor types. In a comprehensive benchmark, PUREE outperformed existing transcriptome-based purity estimation approaches. Overall, PUREE is a highly accurate and versatile method for estimating tumor purity and interrogating tumor heterogeneity from bulk tumor gene expression data, which can complement genomics-based approaches or be used in settings where genomic data is unavailable.

[1] Genome Institute of Singapore (GIS), Agency for Science, Technology and Research (A*STAR), 60 Biopolis Street, Singapore 138672, Republic of Singapore.
[2] School of Computing, National University of Singapore, Computing 1, 13 Computing Drive, Singapore 117417, Republic of Singapore. [3] National Cancer Centre Singapore, Division of Medical Oncology, 30 Hospital Boulevard, Singapore 168583, Republic of Singapore. ✉email: skanderupamj@gis.a-star.edu.sg

Cancerous tumors are complex mixtures of malignant and non-malignant cells shaping the tumor microenvironment (TME). The composition and relative proportions of malignant cells and non-malignant components (comprising stromal, epithelial, and infiltrating immune cells) can display substantial variation across tumors[1–4]. The composition of the TME is also associated with the disease stage and treatment response[5]. The proportion of malignant cancer cells in the tumor mass, herein referred to as tumor purity, also impacts genomic analysis such as the estimation of clonal composition[6] and tumor mutation burden[7], critical for predicting treatment outcomes and selecting patients for immunotherapy. Moreover, tumor purity can guide tumor transcriptome deconvolution and the estimation of gene expression profiles for malignant and non-malignant cell populations inside tumors[8,9], enabling new insights into TME biology[10] and its impact on clinical treatment response.

Traditionally, the cancer cell proportion has been estimated by pathologists inspecting nuclei in hematoxylin and eosin (H&E)-stained tissue slides. However, such estimates may often be imprecise, as demonstrated by the noticeable variation in estimates when the same sample is evaluated by different pathologists[11]. More recent computational approaches to estimate tumor purity are based on DNA sequencing data where variation in allele frequencies of somatic DNA mutations, copy-number alterations (CNAs), or DNA methylation patterns are used to infer the malignant cell proportion[5,12–18]. Genomics-based purity estimation methods, despite differences in underlying statistical models and input data, have been shown to produce concordant estimates of tumor purity[9,19].

Tumor purity can also be estimated from the tumor gene expression profile[20], which has been used to derive clinically relevant molecular subtypes[21–24], perform quality control of tumor samples[25,26], and analyze treatment responses after immunotherapy[27]. Existing methods that estimate tumor purity from a tumor gene expression profile adopt different analytical strategies (Supplementary Note 1, 2). ESTIMATE calculates a combined enrichment score for infiltrating immune and stromal cells followed by training of a supervised model[28]. EPIC uses constrained least square optimization in combination with non-malignant cell-type reference profiles to perform cell-type proportion deconvolution[29,30]. DeMixT uses probabilistic modeling to infer proportions of stromal and cancer-cell components from a set of input samples, comprising both tumor and normal-tissue samples[8]. LinSeed constructs an undirected weighted linearity network of genes to determine mutually linear features followed by simplex-based deconvolution[31]. CIBERSORTx defines a cell-type signature matrix followed by support vector regression to infer the proportions of cell types in each sample[32,33]. Similarly, DeconRNASeq solves a non-negative least squares problem using a pre-defined cell-type signature matrix to derive the cellular proportions[34]. However, due to the inherent modeling assumptions, these methods might not always capture all the biological variation between stroma and cancer cells, required to predict require malignant cells' proportions. Additionally, while these methods show convincing results in their own benchmarks, it is not clear how accurate they are when applied to distinct cancer types and compared with each other.

Our goal was to develop an accurate reference-free method for predicting tumor purity from a tumor gene expression profile. To reduce the modeling limitations of existing approaches, we utilized minimal prior modeling assumptions and instead relied on a statistical learning approach to infer gene expression patterns related to stroma and cancer components. We used a weakly supervised learning strategy, training a machine learning model using gene expression data from 7864 tumors and 20 solid cancer types[35] in combination with orthogonal consensus genomics-based tumor purity estimates. The resulting method, PUREE, is able to robustly predict purity values with high correlation and low root mean squared error (RMSE) when compared to consensus genomics-based estimates from the same samples, outperforming existing deconvolution methods both on a TCGA test set (0.2 increase in Pearson's correlation and 0.17 decrease in RMSE compared to the respective second-best approaches) and seven external validation datasets of the lung, colorectal, uterine, paraganglioma, and testicular cancers.

## Results

**Overview of approach**. Our goal was to develop an accurate method for estimating tumor purity from a tumor gene expression profile. Such a method should be able to generalize across different solid cancer types and exhibit high concordance with orthogonal purity estimates derived from tumor DNA data (Fig. 1a). We therefore assembled a training dataset comprising matched genomic and gene expression profiles from 7864 tumors spanning 20 solid cancer types from TCGA[35] (Supplementary Table 1). The orthogonal (pseudo-ground truth) purity label of each tumor was estimated using the tumor genomic profile, using the consensus of four existing algorithms that generally displayed high concordance (mean Pearson $r = 0.85$, Methods, Supplementary Fig. 1). Next, we adopted a weakly supervised learning strategy to train a model that could predict tumor purity labels from the matched gene expression profiles. Gene expression profiles were rank-percentile transformed to provide robustness to variation in scale and normalization of different gene expression datasets and platforms (e.g. FPKM, TPM; Methods). From the 60,000 transcripts profiled in TCGA, we further selected and focused on 9554 (10 K) highly expressed protein-coding autosomal genes for model development (Methods). We explored the performance of a range of machine learning methods (Supplementary Fig. 2). In particular, given the regression task of predicting the bounded continuous tumor purity value, we tested both a range of linear and non-linear machine learning architectures. This comparison showed that a simple linear regression model could achieve optimal accuracy using only a limited set of gene expression features (Supplementary Fig. 2). PUREE was therefore developed using linear regression and weakly supervised learning strategy to enable accurate estimation of tumor purity from a solid tumor gene expression profile (Fig. 1b).

**Feature selection to account for cancer type and tumor purity imbalance**. The TCGA training dataset showed strong cancer type and purity range imbalance (Fig. 2a, Supplementary Table 1). To reduce the impact of this imbalance during model training, we adopted a two-step feature selection strategy (Methods). Briefly, the first step consisted of selecting features that could predict purity at both lower and higher purity ranges (Fig. 2b). The second step further filtered this feature set to identify the genes most predictive across the entire purity range, resulting in 158 features (Fig. 2c).

We further explored the properties of the resulting reduced feature set. Using gene set enrichment analysis, the 158 genes were enriched in pathways relating to angiogenesis, KRAS signaling and epithelial-mesenchymal transition (Methods, Supplementary Fig. 3). Genes positively correlated with purity showed enrichment in cancer-related pathways and processes such as epithelial-mesenchymal transition (*BASP1*, *COL4A1*, *THBS2*), genes involved in the TNFA signaling via NFKB (*SPSB1*, *SMAD3*), and genes upregulated by KRAS activation (*CFB*, *MAFB*). Genes negatively correlated with purity showed enrichment in stroma-related processes such as inflammatory response (*IL1R1*, *STAB1*, *MSR1*) and also genes involved in epithelial-

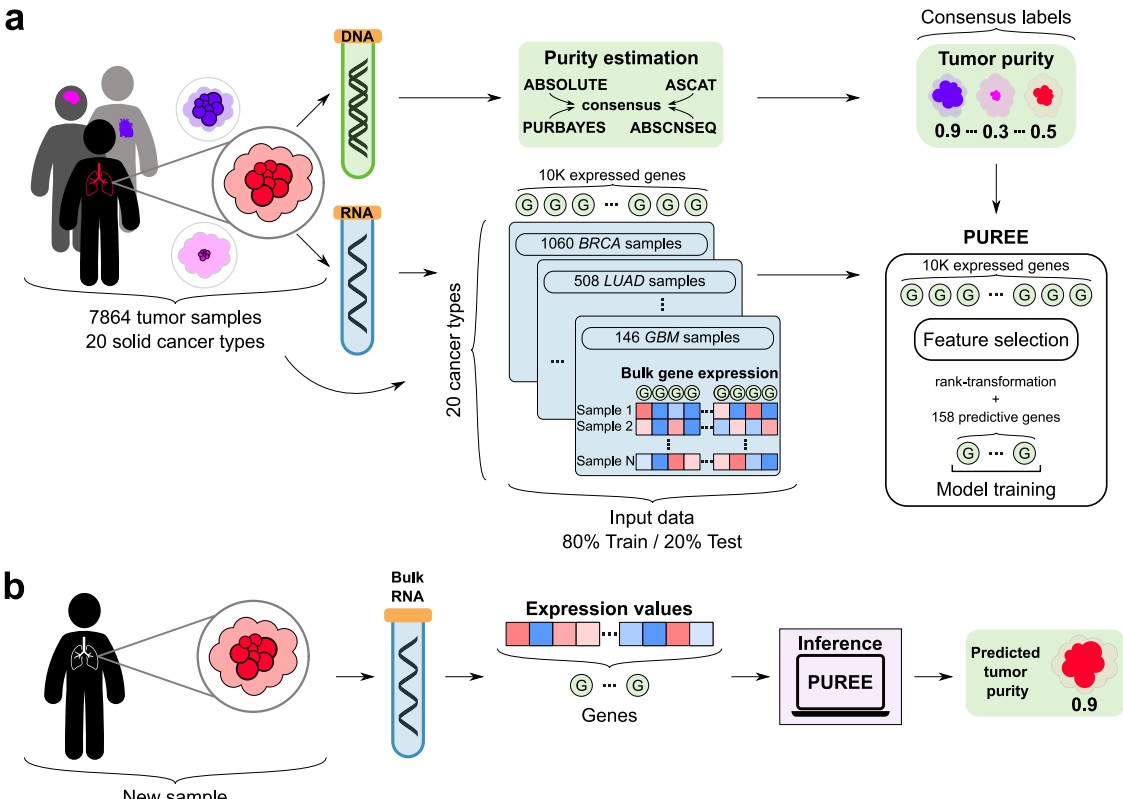

**Fig. 1 Overview of PUREE. a** PUREE is trained using a weakly supervised learning approach. Consensus genomics-based purity estimates are used as orthogonal (pseudo-ground-truth) labels, and a predictive model is trained on rank-transformed gene expression profiles from 7864 tumor samples spanning 20 solid tumor types (80%/20% train/test split). **b** For a new solid tumor sample, PUREE infers the purity from the corresponding tumor gene expression profile.

mesenchymal transition (*TGFBR3, CXCL12, CRLF1, PMP22, SDC1*). Overall, this confirmed the original hypothesis of the PUREE model selecting cancer and stroma-related genes.

**Comparing the performance of pan-cancer and cancer-type-specific models**. As an alternative to PUREE's pan-cancer tumor purity prediction model, we explored whether models trained for a specific cancer type could more accurately predict purity. To test this, we trained cancer-type-specific models (Methods) and compared their performance with PUREE across all cancer types. Interestingly, PUREE showed comparable and often improved performance, with comparable median correlation (0.784 vs 0.790, $P = 0.1$, Wilcoxon signed-rank test, two-tailed) and lower median RMSE (0.094 vs 0.096, $P = 0.08$) with the orthogonal genomics-based purity estimates (Fig. 3a, Supplementary Figs. 4–6). Overall, this confirmed that PUREE's pan-cancer feature selection and training approach provided a robust and accurate prediction across all individual cancer types.

Next, we evaluated the ability of PUREE to predict purity in cancer types absent from the training data (Methods). We compared performance metrics for PUREE and versions of PUREE where one cancer type was removed from the training data. This comparison showed only a minor decrease in correlation (median 0.7847 vs 0.7843, $P = 0.0005$, Wilcoxon signed-rank test, two-tailed) and an increase in RMSE (0.094 vs 0.099, $P = 6e{-}6$) when the cancer type was absent from the training data (Fig. 3b, Supplementary Fig. 7). This demonstrated that PUREE is robust and can generalize to solid tumor types not included in the training data, and that using the reduced feature set provides more robustness to the model.

**Benchmarking of methods on independent datasets**. We evaluated PUREE's performance on the withheld test sets from the TCGA dataset. We compared PUREE with six existing transcriptomics-based deconvolution and purity estimation methods (Methods).

Here, PUREE consistently demonstrated higher correlation and lower RMSE with consensus purity labels than the existing deconvolution methods (Fig. 4, Supplementary Fig. 8). PUREE had the highest median correlation ($r = 0.78$), followed by ESTIMATE (0.63) and CIBERSORTx (0.55). Similarly, PUREE had the lowest median RMSE of all methods (0.09), 53% lower than the next-best method (CIBERSORTx, 0.19), and PUREE displayed the lowest RMSE in each cancer type. PUREE also showed less variation in performance across cancer types as compared to the other methods, with an inter-quartile range for correlation and RMSE of 0.12 and 0.015, respectively. We additionally evaluated the performance of methods when tested across solid tumor types with likely distinct stromal composition (e.g. brain cancers and skin cancers). Consistent with our previous observations, PUREE outperformed other transcriptomics-based approaches, showing comparable high accuracy across cancer types with expected dissimilar stromal composition (Supplementary Fig. 9). A similar analysis showed that PUREE outperformed the other methods on the cancer types with extreme median tumor purities (Supplementary Fig. 10).

Next, we compared PUREE's and other methods' performance on two additional independent public lung cancer cohorts[36,37], a colorectal cancer cohort[38], and 4 TCGA cohorts of colorectal, uterine endometrial, pheochromocytoma and paraganglioma, and testicular cancer not present in the initial TCGA dataset used for model training and testing

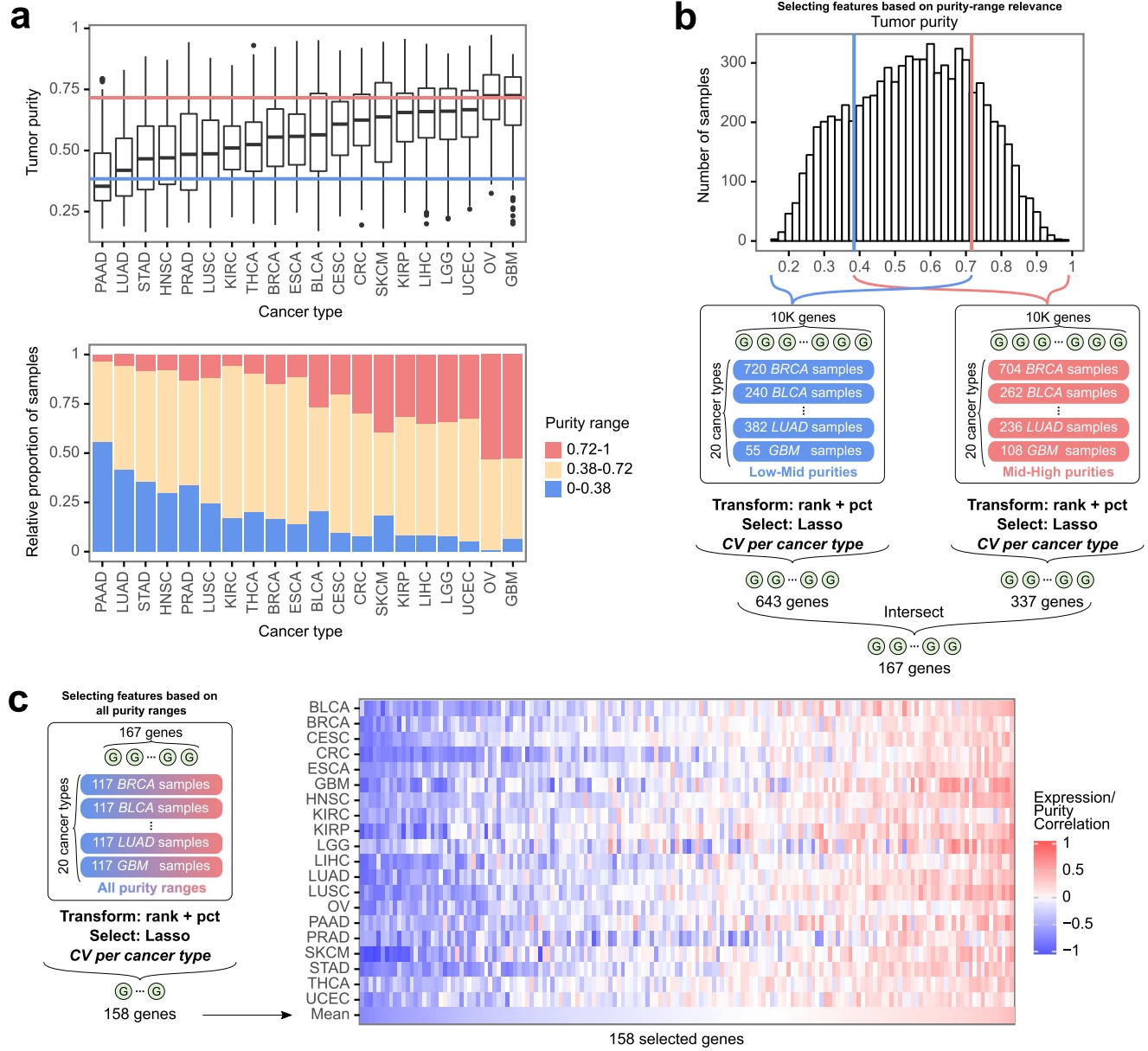

**Fig. 2 Feature selection to account for cancer type and tumor purity imbalance. a** Boxplots of the genomics-based tumor purity across 20 cancer types from TCGA, blue and red lines marking the bottom/top 20% purity samples; bar plots of the genomics-based tumor purity separated into low, medium, and high purity ranges. **b** The first step of the feature selection strategy: lasso feature selection, cross-validated on cancer types as folds, was performed separately on low-mid (0.17–0.72 purities) and mid-high (0.38–0.97) purity range samples, the two feature sets were intersected resulting in 167 genes. **c** The second step of the feature selection strategy: using the features from the first step, lasso feature selection was iteratively performed across all purity ranges, cross-validated on cancer types as folds, resulting in 158 gene features used for the final model. In the boxplots in (**a**), the lower and upper hinges correspond to the first and third quartiles, the upper whisker extends to the largest value no further than 1.5 of inter-quartile range from the hinge, the lower whisker extends to the smallest value no further than 1.5 of inter-quartile range from the hinge, and points beyond the end of the whiskers are plotted individually.

(Supplementary Table 2, Methods). Similar to the TCGA cohort, orthogonal genomics-based tumor purity estimates in these cohorts were estimated from tumor DNA sequencing data (Methods). Across all seven cohorts, PUREE demonstrated generally higher correlation and lower RMSE with the genomics-based tumor purity estimates (Fig. 5, Supplementary Fig. 11). We additionally compared PUREE's resource usage against the other methods in terms of memory (RAM) and compute time. This evaluation showed that PUREE uses less memory to run, in addition to consistently being the fastest method (Supplementary Fig. 12).

**Exploring the PUREE feature set using single-cell RNA-seq data.** We performed orthogonal analysis and validation of the 158 gene features in the PUREE model using single-cell RNA-seq data. We used published scRNA-seq data from head and neck cancer[39] (5902 cells total, 2539 classified as malignant, 3363 as non-malignant) and melanoma[40] (4513 cells total, 3256 classified as malignant, 1257 as non-malignant). We computed mean cell-wise z-scores of the expression of the genes with positive and negative purity-expression correlation. Interestingly, these genes showed noticeable expression differences between malignant and non-malignant cells in both tumor types (Fig. 6, Mann–Whitney $P < 1e-90$; Supplementary

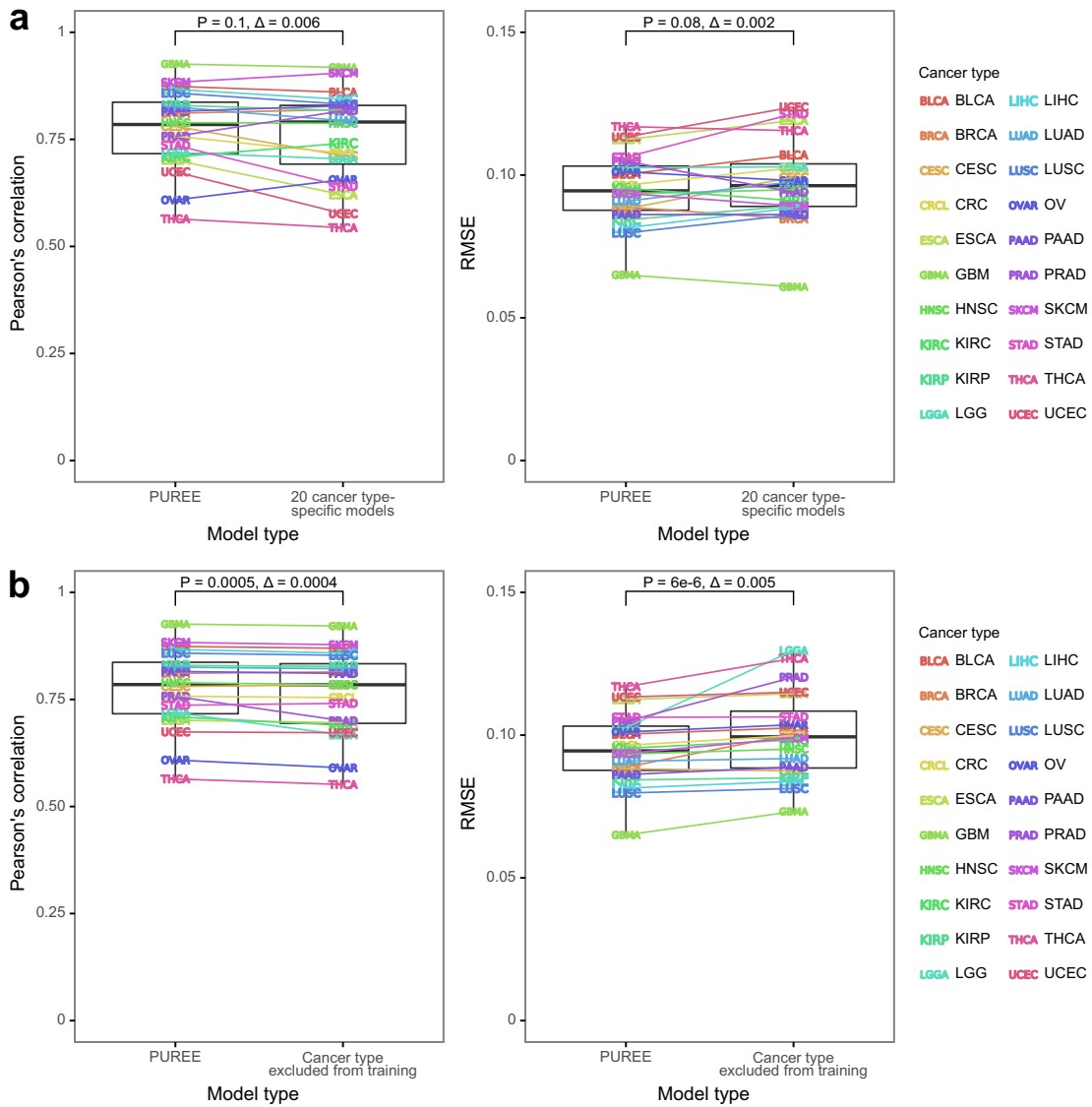

**Fig. 3 PUREE versus cancer type-specific architectures and unseen cancer types. a** Performance (Pearson's correlation and RMSE) of PUREE as compared to 20 cancer-type-specific models on the test split of the TCGA dataset. **b** Performance (Pearson's correlation and RMSE) of PUREE when predicting on a cancer type included in training data versus when the same cancer type was excluded from the training data. In the boxplots, the lower and upper hinges correspond to the first and third quartiles, the upper whisker extends to the largest value no further than 1.5 of inter-quartile range from the hinge, the lower whisker extends to the smallest value no further than 1.5 of inter-quartile range from the hinge, and points beyond the end of the whiskers are plotted individually. Differences were evaluated with the Wilcoxon signed-rank test (two-tailed), and delta is the difference between medians of metrics across all cancer types. P-values from Wilcoxon signed-rank test (two-tailed) shown.

Fig. 13). Genes with positive purity-expression correlation had markedly higher expression in malignant cells as compared to non-malignant cells. In contrast, genes with negative purity-expression correlation were upregulated in non-malignant cells. This result further confirmed that the gene feature set used by PUREE has the ability to distinguish between and quantify the proportion of cancer and non-cancer cells in the tumors.

## Discussion

We developed a computational method, PUREE, that can predict the proportion of cancer cells in a sample (tumor purity) from a bulk tumor gene expression profile. The method adopts a machine learning-based feature selection strategy in combination with a linear regression architecture. The model is trained using weak supervision and consensus tumor purity labels obtained from tumor DNA sequencing data spanning 20 solid tumor types and a range of median genomics-based purities from 35% in pancreatic cancer to 72% in ovarian cancer. Compared with existing transcriptome deconvolution methods, we demonstrate that PUREE has superior accuracy across multiple independent test cohorts, spanning median purity ranges from 45% to 76%. Additionally, PUREE is fast and user-friendly as the underlying model is pre-trained in advance.

PUREE adopts a stringent feature selection strategy, with which we were able to reduce the initial feature set of 9554 genes to 158 predictive genes. Combined with a pan-cancer training strategy, we demonstrate that this sparse model can generalize well to unseen tumor types and cohorts. We also show that PUREE's pan-cancer architecture has comparable accuracy to cancer-type-specific models. This suggests that the model is able to capture expression signatures of cancer and stromal cells

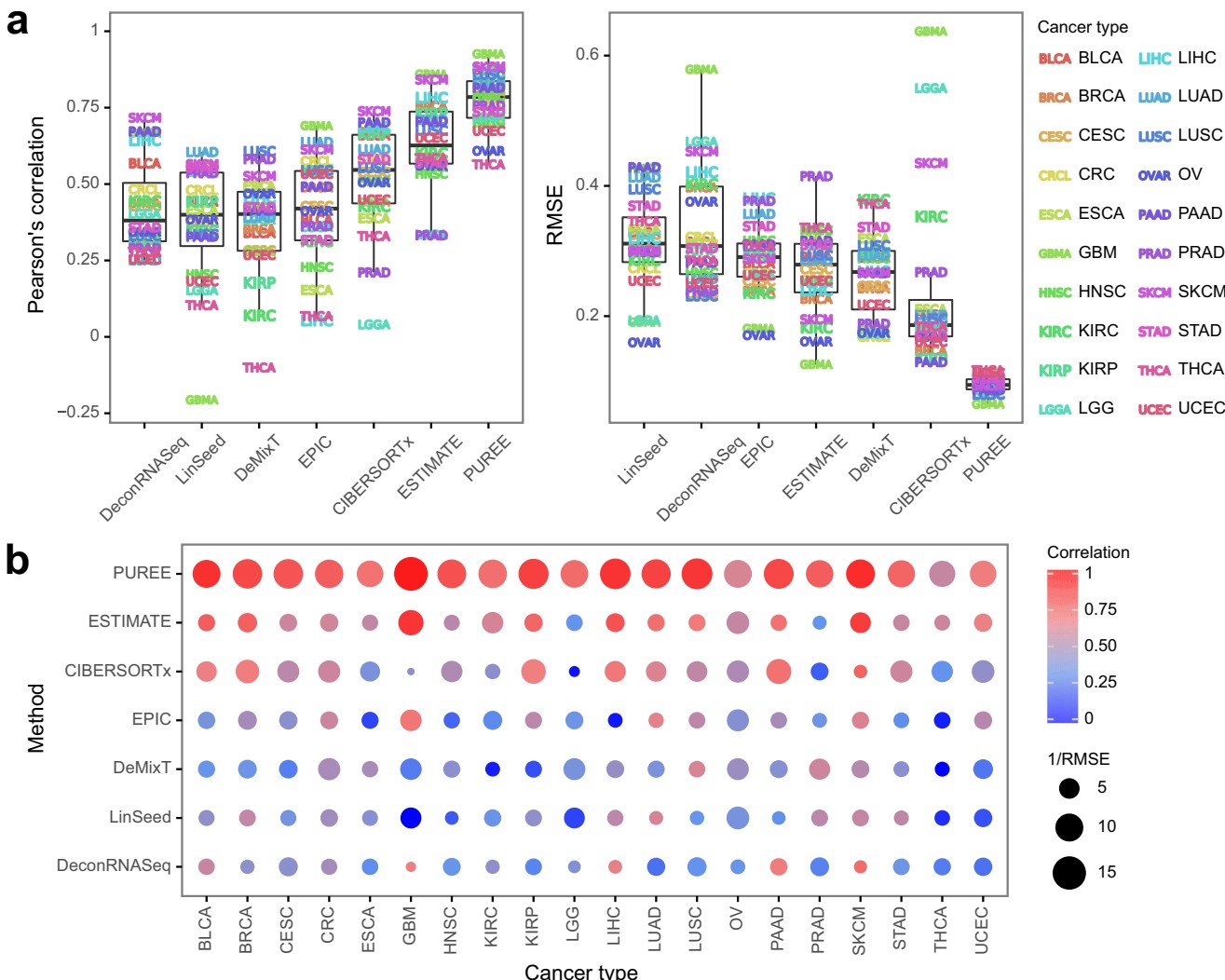

**Fig. 4 Comparing PUREE and existing methods on TCGA data.** Comparison of PUREE and 6 existing transcriptome deconvolution methods. **a** Mean Pearson's correlation and RMSE of methods per cancer type when compared with genomic tumor purity estimates on the TCGA test data split (1573 samples). **b** Aggregated mean correlation and RMSE metrics per cancer type for each method. In the boxplots in (**a**), the lower and upper hinges correspond to the first and third quartiles, the upper whisker extends to the largest value no further than 1.5 of inter-quartile range from the hinge, the lower whisker extends to the smallest value no further than 1.5 of inter-quartile range from the hinge, and points beyond the end of the whiskers are plotted individually.

conserved across solid tumor types and that cancer-type-specific expression signatures do not provide additional discriminatory information for tumor purity estimation. We found that these pan-cancer conserved feature genes were enriched for known cancer and stromal cell-specific processes such as epithelial-mesenchymal transition and immune cell activity.

The gene feature set used by PUREE also demonstrated a remarkable ability to distinguish between malignant and non-malignant cells in single-cell RNA-seq data. This provides orthogonal validation of our feature selection and pan-cancer training strategy, and further confirms the predictive power of the selected gene set. This result also suggests that a modified version of our approach could potentially be repurposed to classify malignant and non-malignant cells in single-cell RNA-seq data. Finally, due to its supervised machine learning approach, PUREE has some limitations stemming from the composition of the training data. Specifically, the method has only been trained and tested on solid tumor samples and will therefore likely have suboptimal performance if applied to other non-solid cancer types.

In summary, we have shown that PUREE is a highly accurate and efficient method for purity estimation from a tumor gene

expression profile, enabling robust and accurate interrogation of tumor purity and heterogeneity from bulk tumor gene expression data. We envision PUREE to be especially useful in settings where the DNA-seq data is either hard to obtain or absent. Even when tumor DNA-seq data are available, PUREE may provide an additional and orthogonal approach to tumor purity estimation. This may be especially relevant in cohorts and settings where the DNA and RNA are extracted from different aliquots of a tumor.

## Methods

**Genomics-based consensus tumor purity estimates.** For TCGA samples, genomic-based consensus tumor purities were computed as a mean of predictions from ABSOLUTE[17], AbsCNSeq[18], ASCAT[15], and PurBayes[16] following the approach reported in Ghoshdastider et al. [41]. AbsCNSeq and PurBayes estimates are based on mutation variant allele frequency data, and ASCAT and ABSOLUTE on SNP-array data. Briefly, samples with extremely low (<0.1) and extremely high (>0.98) purity estimates from individual methods were flagged as missing data, as recommended by Ghoshdastider et al. These missing data values were instead imputed using an iterative principal component analysis approach[42]. Quantile normalization was used to standardize and average the tumor purity distributions of different algorithms per cancer type. Finally, consensus purity estimates were estimated as the sample-wise medians of the normalized purity estimates from individual methods.

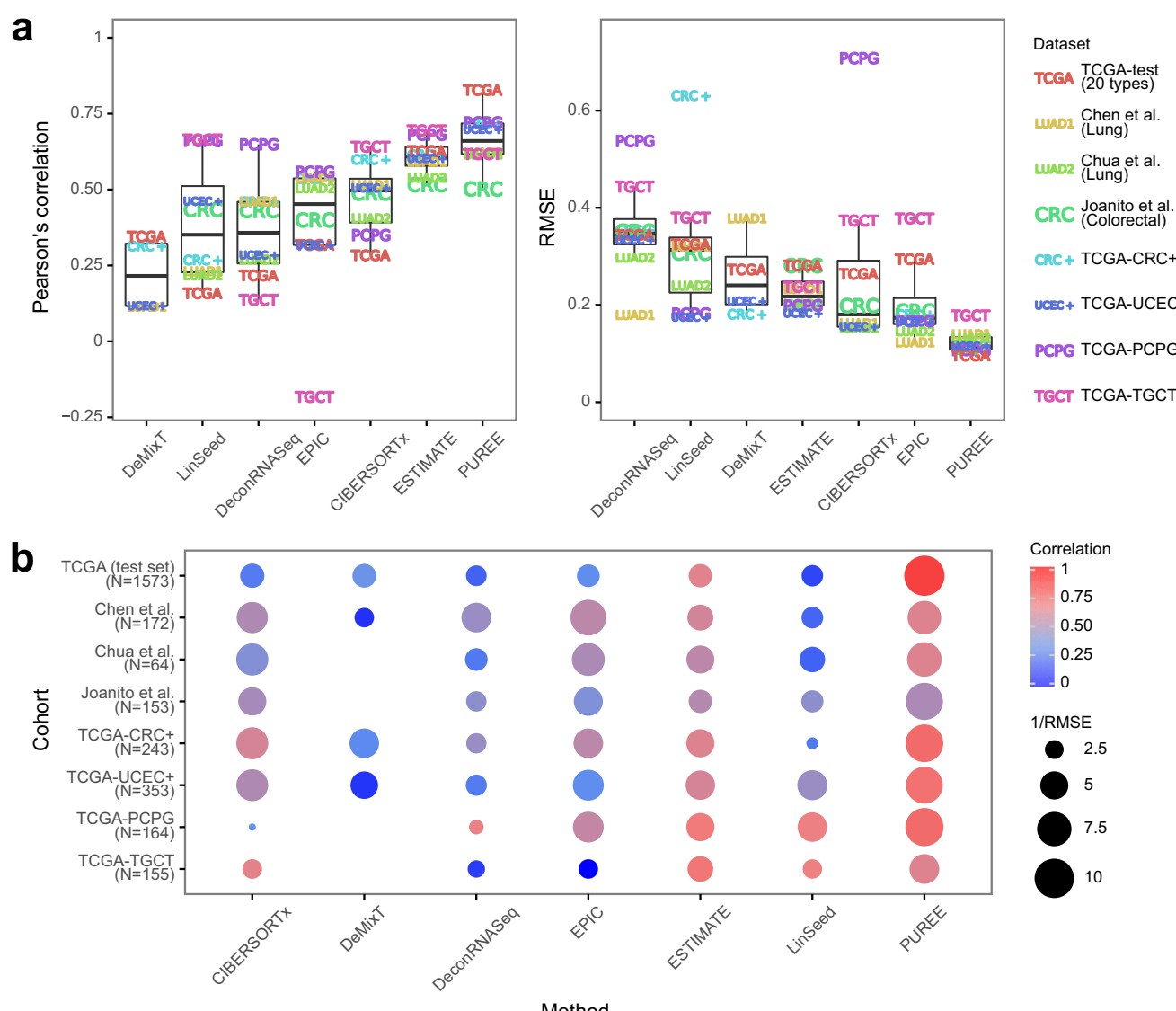

**Fig. 5 Benchmarking of methods on independent datasets.** Comparing the purity estimates of PUREE and 6 existing transcriptome deconvolution methods. The predictions from each method were compared with genomic tumor purity estimates across 8 cohorts: the TCGA test set, lung cancer samples from Chen et al. [36], lung cancer samples from Chua et al. [37], a colorectal cancer cohort from Joanito et al. [38], and 4 TCGA cohorts of colorectal (CRC+), uterine endometrial (UCEC+), pheochromocytoma and paraganglioma (PCPG), and testicular cancer (TGCT) not present in the initial TCGA dataset used for training and testing. **a** Mean Pearson's correlation and RMSE per cohort for all methods. **b** Mean correlation and RMSE per cohort of every method shown together. DeMixT could not be run on the Chua et al., Joanito et al., PCPG and TGCT cohorts due to the absence of normal samples there. In the boxplots in **a**, the lower and upper hinges correspond to the first and third quartiles, the upper whisker extends to the largest value no further than 1.5 of inter-quartile range from the hinge, the lower whisker extends to the smallest value no further than 1.5 of inter-quartile range from the hinge, and points beyond the end of the whiskers are plotted individually.

**TCGA training and test set construction**. The TCGA dataset consisted of 7864 samples from 20 solid cancer types. 80% of samples were selected for model training (TCGA train split, 6291 samples) and 20% for testing (TCGA test split, 1573 samples). The training and test sets were randomly sampled so they had comparable cancer-type and purity distributions. The initial gene expression feature matrix was filtered to only include autosomal and protein-coding genes. Genes with low expression (median TPM < 1) and low variance (variance < 1) in all cancer types in the TCGA train split were further filtered, leaving 9554 gene expression features for subsequent steps (referred to as the 10 K features set).

**Gene expression data rank-transformation**. Gene expression data is rank-percentile normalized (sample-wise) when serving as input for PUREE. The initial rank-transformation allows for generalization across different gene expression platforms (e.g. RNA-seq, microarray) and measurement units (e.g. Transcripts Per Million (TPM), Fragments Per Kilobase Million (FPKM)). Briefly, gene expression values (e.g. TPMs [0, 0, 1, 5, 100]) are first ranked based on their position within a sample in ascending order, assigning the lowest possible rank for tied groups

([1, 1, 3, 4, 5]). The percentile of the resulting rank is then computed ([0.2, 0.2, 0.6, 0.8, 1]). The resulting percentiles computed relative to the ranking universe of the 10 K feature set serve as input values to PUREE.

**Construction of machine learning models**. All machine learning models (Elastic Net, Gradient Boosting, nu-Support Vector Regression, Lasso, Logit Regression and Linear Regression models, as well as Simple Imputer for missing values imputation) were constructed and trained using the Scikit-Learn Python package[43]. Logit Regression was built as a modified version of Linear Regression from Scikit-Learn. A fully connected Neural Net consisting of a variable number of relu-activated fully connected hidden layers, depending on the feature size of the input data, was constructed and trained using the Keras submodule of the Tensorflow Python package[44]. All the hyperparameters not explicitly defined in the model call functions (e.g. alphas for Lasso) or in the hyperparameter search functions (e.g. HalvingGridSearchCV), were allowed to either be chosen by the in-built hyperparameter selection procedure or be used at their default values.

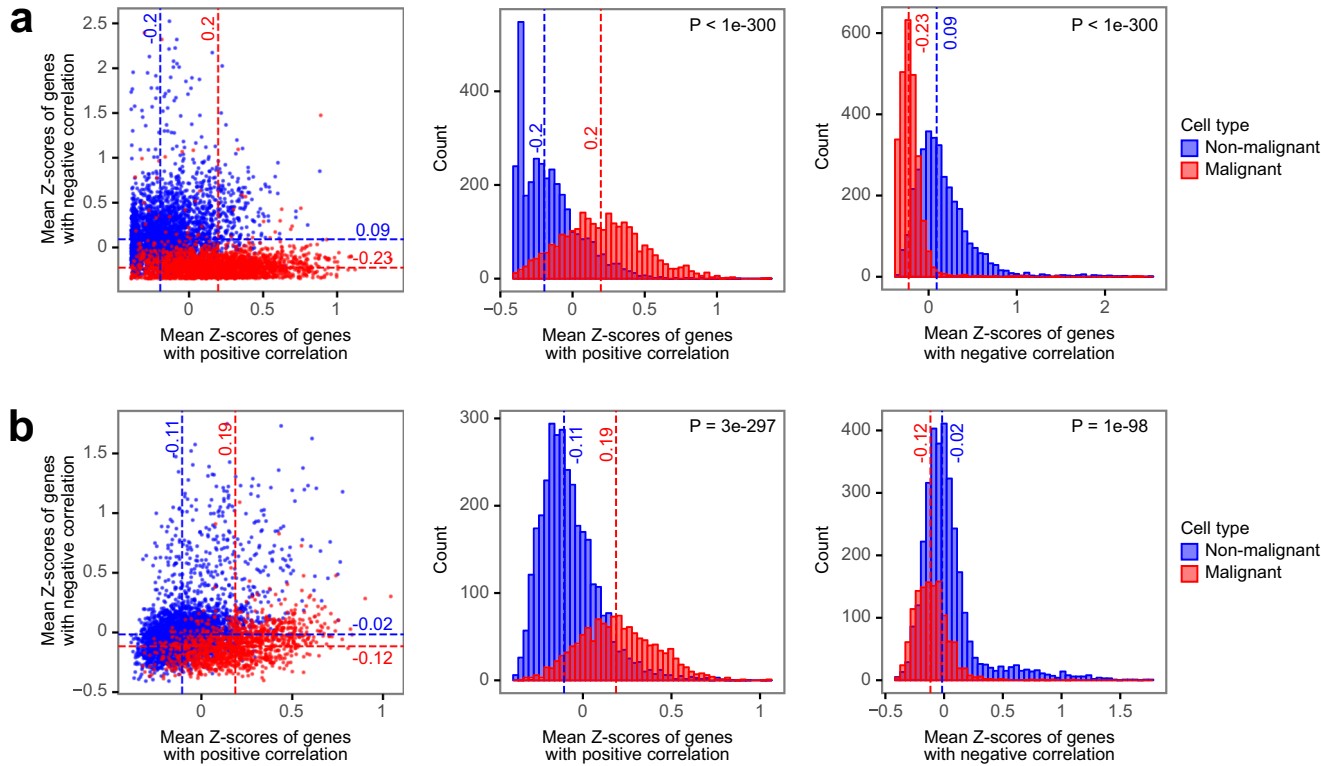

**Fig. 6 Validation of PUREE's features using single-cell RNA-seq data. a, b** Comparing the expression of PUREE gene features in malignant and non-malignant cells in (**a**) head and neck cancer cells from Puram et al. [39] and (**b**) melanoma cells from Tirosh et al. [40]. Malignant and non-malignant cell labels were obtained from the original publications. Genes were separated into groups of positive (top 20%) and negative (bottom 20%) purity-vs-expression correlation in TCGA data averaged across cancer types. The dashed red and blue lines indicate medians of malignant and non-malignant cells' z-scores distributions respectively. *P*-values are calculated using the Mann–Whitney *U* rank test (two-tailed).

**Feature selection**. We used Lasso regression to reduce the number of input features in PUREE. During the first step, we selected features relevant to both low and high purity ranges. As feature selection models, two lasso regression models, cross-validated using cancer types as folds, were trained on all training data except the bottom and top-20% purity values, respectively. These two resulting feature sets were intersected, resulting in 167 genes. Next, we further iteratively selected features equally relevant to all cancer types by training a Lasso model (cross-validated with cancer type as folds) on a balanced subset of the full TCGA training set comprising all purity ranges, using the earlier selected 167 genes as initial features. Briefly, this is done by training on samples from $N-1$ cancer types, and testing on samples from the remaining withheld cancer type (leave-one-cancer-type out). We perform balanced feature selection by selecting 117 samples (determined by tumor type with the lowest number of samples in the training set: GBM, $N = 117$) from each cancer type while preserving the original purity distribution. This resulted in 158 genes with non-zero weights, which serve as predictive features in the final PUREE model. Correlations of the genes' expression with genomic-based tumor purity were computed on the train portion of TCGA as means per each cancer type averaged across all cancer types.

**Predicting on unseen samples**. The PUREE pipeline consists of three parts: rank-percentile normalization, missing values imputation and linear regression model inference. During the first step, rank percentiles of the overlap of the genes in the input data and the 10 K genes are computed. The data is further reduced to the 158 PUREE input genes. During the second step, if there are any missing values in the 158 selected genes, they are imputed based on the medians of the values in TCGA train set. During the third step, the pre-trained linear regression model is applied to the resulting data in order to predict purity values. The predicted values that fall out of the [0,1] range are rounded to the nearest in-range value.

**Cancer-type-specific models and test experiments**. For experiments with cancer type-specific models, we constructed training/test sets comprising the 10 K features for each individual cancer type. We used 5-fold cross-validation to train lasso models on these cancer-type-specific training sets. For experiments where individual cancer types were excluded during model training, we constructed a pan-cancer training set (158 features) comprising all cancer types except the cancer type being withheld, followed by linear regression training as described for the main pan-cancer PUREE model.

**Gene set enrichment analysis**. Gene set enrichment analysis was evaluated using the GSEApy Python package (https://github.com/zqfang/GSEApy), which is based on Enrichr[45] and Gene Set Enrichment Analysis (GSEA)[46]. The names of 158 selected genes were converted into HGNC nomenclature and used as an input to the enrichr function. Background genes were set to be 9554 significantly expressed autosomal genes in TCGA. The hallmark gene set used in the enrichment analysis was downloaded from the MSigDB collection (http://www.gsea-msigdb.org/gsea/msigdb/collections.jsp, set H). For each gene, Pearson correlation was computed between its expression and DNA-based tumor purity in each of the 20 cancer types in the TCGA train set, and a mean of it was taken. Top 10 enriched pathways by Benjamini-Hochberg adjusted p-value were computed for 4 gene sets: full feature set of 158 genes, top 30% genes by their mean expression-purity correlation per cancer type, bottom 30% genes by their mean expression-purity correlation per cancer type, and, finally, the genes in the 30-70% range by their mean expression-purity correlation per cancer type.

**Running other transcriptomics-based methods for purity prediction**. Unless explicitly stated otherwise, the packages below were run in R environment version $> = 3.4$. For all the gene expression matrices below, only protein-coding genes were left for the downstream analysis. The gene ids were used in HGNC nomenclature.

CIBERSORTx[32] was run using the web interface available at https://cibersortx.stanford.edu/. The analysis module was selected to be "Impute Cell Fractions", "Custom" mode and "RNA-seq" input data. The NSCLC signature matrix used for imputation was taken from CIBERSORTx's paper[32] supplementary 2 l. Mixture files were used in linear space (in TPM values when available, otherwise FPKM) and formatted according to the instructions provided on the website. Batch correction was run in B-mode with no GEP. Quantile normalization was disabled. 100 permutations were used for statistical analysis. "EPCAM" column was taken as tumor purity.

DeMixT[8] (https://github.com/wwylab/DeMixT) was run on gene expression matrices in linear counts space. As the DeMixT package required tumor and normal counts (not necessarily matched), it was run only on datasets that had both available. Additionally, the counts matrices were quartile-normalized and the genes where the total sum of values across all samples was <1 were discarded. As DeMixT seemed to predict the stromal component, the purity was computed as (1-DeMixT predictions).

EPIC[29,30] (https://github.com/GfellerLab/EPIC) was run on gene expression matrices in linear normalized values space (TPM or FPKM). Purity was taken as the 'otherCells' component of the resulting cellFractions.

ESTIMATE[28] (https://rdrr.io/rforge/estimate/) was run on gene expression matrices in linear normalized values space (TPM or FPKM). For the estimateScore function the platform parameter was chosen to be "affymetrix".

LinSeed[31] (https://github.com/ctlab/LinSeed) was run on gene expression matrices in linear values space (counts if they were available, or TPM or FPKM if not). Additionally, the matrices were normalized sample-wise so that the samples would have the same sum. RPL/RPS genes were removed from the matrix. LinseedObject function was run with topGenes = 10,000, the rest of the functions were run in a 2-component mode according to the instructions provided by the authors in their GitHub repository https://github.com/ctlab/LinSeed. As LinSeed does not explicitly state which of the deconvolved components represents the cancer cells' proportion, the component that had the best Pearson's correlation with DNA-based purities was taken. Additionally, since it appeared that sometimes LinSeed might be predicting the stromal cell proportion instead of purity, if the 1- (LinSeed predictions) had better correlations, that was taken as predicted purity instead.

DeconRNASeq[34] (https://doi.org/10.18129/B9.bioc.DeconRNASeq) was run on gene expression matrices in linear normalized values space (TPM or FPKM), CIBERSORTx's NSCLC matrix was used as a signature. The "EPCAM" column was taken as tumor purity.

**Statistics and reproducibility.** To calculate P-values, Mann–Whitney U rank test was used for non-paired data (single-cell z-scores) and Wilcoxon signed-rank test for paired samples (cancer types). Pearson's correlation and root mean squared error were used to calculate the mean statistics for each cancer type or cohort.

TCGA cohort of 20 cancer types includes 7864 samples with 6291 in the train set and 1573 in the test set. Chen et al. lung cancer cohort consists of 172 samples. Chua et al. lung cancer cohort consists of 64 samples. Joanito et al. colorectal cancer cohort consists of 153 samples. TCGA-CRC+ colorectal cancer cohort consists of 243 samples. TCGA-UCEC+ uterine endometrial cancer cohort consists of 353 samples. TCGA-PCPG pheochromocytoma and paraganglioma cohort consists of 164 samples. TCGA-TGCT cohort consists of 155 samples.

The head and neck single-cell RNA-seq cohort from Puram et al. consists of 5902 cells, 3363 of which are malignant and 2539 non-malignant. The melanoma single-cell dataset from Tirosh et al. consists of 4513 cells, 3256 of which are malignant and 1257 non-malignant (132 cells with unresolved cell type assignment were dropped).

**Reporting summary.** Further information on research design is available in the Nature Portfolio Reporting Summary linked to this article.

## Data availability

TCGA gene expression data on 20 solid cancer types were downloaded through the UCSC Xena Hub (https://xenabrowser.net/datapages/)[47]. The gene expression data of the lung cancer validation cohorts used in the external benchmark were obtained from the publication by Chen et al. [36] and Chua et al. [37]. The gene expression data of the colorectal cancer cohort used in the external benchmark was obtained from the publication by Joanito et al. [38]. The preprocesed gene expression data of 4 TCGA cohorts of colorectal, uterine, paraganglioma and testicular cancers were also obtained from the UCSC Xena Hub[47]. The gene expression data and the cell type labels of the head and neck cancer single-cell cohort used in the feature set validation was taken from the publication by Tirosh et al. (Gene Expression Omnibus ID GSE72056)[40]. The gene expression data and the cell type labels melanoma single-cell cohort used in the feature set validation were obtained from the publication by Puram et al. (Gene Expression Omnibus ID GSE103322)[39]. In the first lung cancer validation dataset from Chen et al. [36], genomic purity estimates were originally computed as a mean of THetA2[48], TitanCNA[49], AbsCNSeq[18] and PurBayes[16] and obtained from the respective publication. In the second lung cancer validation dataset, Chua et al. [37], genomic purity estimates were originally based on ASCAT[15] and Sequenza[50] methods and taken from the respective publication. In the colorectal cancer validation cohort from Juanito et al., genomic consensus purity values were re-computed as a mean of THetA2[48], TitanCNA[49], AbsCNSeq[18] and PurBayes[16] methods. ABSOLUTE tumor purity estimates for TCGA samples were obtained from the NCI Genomic Data Commons (GDC) database[35]. Only samples that had genomic tumor purity available were used. All public datasets used in this study were collected with appropriate ethical approvals. The source data to generate the figures were deposited to Zenodo (https://doi.org/10.5281/zenodo.7772812).

## Code availability

PUREE is available as a web service (https://puree.genome.sg/) and the respective Python package (https://github.com/skandlab/PUREE). The version of the code for the PUREE package used to generate the data for the publication was deposited to Zenodo (https://doi.org/10.5281/zenodo.7772812)[51]. The source data and the codes used to

generate the figures, as well as the codes used to conduct the methods benchmark and to set up the machine learning models were also deposited to Zenodo (https://doi.org/10.5281/zenodo.7772812).

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

## Acknowledgements

During the method's development, E.R. was a student funded by the SINGA-A*STAR scholarship and is grateful for the financial support provided for the research above. The authors would also like to express gratitude to Sim Ngak Leng for the help with relevant bioinformatics analysis, Kiran Krishnamachari for the extensive discussion on the manuscript, Sinem Kadioglu and Yu Amanda Guo for testing the method, and members of the Skanderup lab for other helpful inputs and overall support of the project. This research is supported by the Singapore Ministry of Health's National Medical Research Council under its OF-IRG program (OFIRG18may-0075), and Agency for Science, Technology and Research (A*STAR) under its CDAP program (grant no. 1727600057).

## Author contributions

A.J.S. and E.R. conceived the idea and the experiments, E.R. conducted the experiments, developed the method, and wrote the manuscript. T.K. developed the web service and the Python API package. A.J.S. and K.W.-K.S. supervised the method development and reviewed the manuscript.

## Competing interests

The authors declare no competing interests.
