## [Peer Review File · Communications Biology]

Reviewers' comments:

Reviewer #1 (Remarks to the Author):

PUREE is a weakly supervised machine algorithm to infer tumor purity from a tumor gene expression profile. Using LASSO regression, the algorithm first selects the most relevant features (genes) to be used in a pan-cancer model able to predict tumor purity. Trained on a set of 8000 TCGA samples for which genomic estimates of tumor purity are available, PUREE was the best performer among 5 different deconvolution algorithms tested and also on scRNA-Seq data.

While the methodology used is appropriate, the impact and novelty of the method is limited since many other methods are already available for this task, both from an algorithmic and technical perspective. For example, why using expression data to infer tumor purity if genomic data gives you a more straight forward answer for the same question? Performing RNA-Seq is still a relatively expensive method that doesn't work well for samples conserved for example in parafine blocks.

On the other hand, the authors only compare to methods based on bulk RNA-Seq. In the last years, deconvolution methods based on scRNA-Seq have been shown to outperform the methods used in the paper for benchmarking. These scRNA-Seq deconvolution methods can estimate not only tumor purity but also the percentage of each cellular type present in a tumor sample, allowing for instance quantification of tumor infiltrated lymphocytes and their type, what is highly relevant for therapeutic handling of the patients. Some of these methods such as MUSIC, Bisque and DECONRNASeq are summarized in recent reviews (<https://www.nature.com/articles/s41467-022-28655-4>, <https://www.nature.com/articles/s43018-022-00356-3>) and would need to be included in the comparison.

Reviewer #2 (Remarks to the Author):

The manuscript by Revkov et al. entitled "Accurate pan-cancer tumor purity estimation from gene expression data" uses a weakly supervised learning approach to infer tumor purity using gene expression profiles. Their method (PUREE) was used in 8000 bulk samples and single-cell datasets.

Comments:

Page 2, second paragraph: Why the pathologist variation? Can you provide an estimate of the intra- and between observer variability? if available.

Page 2, second paragraph: you forgot to mention the use of DNA methylation methods such as InfiniumPurify, MethylCIBERSORT, etc.

Page 2 end of page 4: high correlation and low RMSE vs. what standard? Here you mention the silver standard; is this the CPE? this needs more explanation here.

Suppl Fig S1: If possible you should signal which methods are DNA-based vs RNA-based. Also please include the n for the training and testing.

Results: The actual number is 7864 tumors? You should use that in your abstract or explain why some samples were excluded. Similarly, use the actual number of transcripts; is it 10K or 10037? Please be concordant through the text.

Pages 4 and 5: Could you check the order of your supplementary figures? Why does S2 appear in the text after S3?

Page 4, supply Fig. S3: you should include some numbers about the selection of Lasso as your final

model. What is the magnitude of the difference vs other methods? what is the range of errors in your estimates?

Figure 2, page 5-6: The selection of the targets based on the extreme purities is very smart. However, the question is whether the lower purity samples are homogeneous across different cancer types? What happened in tumors with very dissimilar stroma (SKCM, PAAD, GBM, LGG, or UCEC) is this affecting the estimates?

Page 7, and Fig S8: Although CPE is pretty commonly used as a silver standard, some authors consider that the ESTIMATE could be a better proxy. Could you briefly expand on why that is the case?

Figure 3: The colors are very hard to read, and probably they are not color blind friendly. I suggest that you use a different palette. In addition to that can you expand on why SKCM is better in cancer-specific models with higher correlation and lower RSME? Something similar happens with STAD, but it is hard to distinguish with the colors. Please explain why that is the case to the readers.

Figure 6: Could you expand on the differences in the densities between those genes with positive and negative correlations with malignant cells? Could you also add the actual numbers?

Page 12: are there any essential differences between cancer types that you should mention in this context? What about the differences with DNA-based method estimates?

Reviewer #3 (Remarks to the Author):

The authors present a manuscript on a machine learning approach to estimate purity from a bulk tumor gene expression profile. The tool is based on a lasso regression classifier over 170 features derived by a stringent feature selection strategy and trained over 7864 TCGA solid tumors. The manuscript is concise, generally clear, and the results appear convincing.

I have, however, a few concerns.

1. The authors should clarify why only 7864 TCGA tumors were used and how they selected these tumors. Also, the distribution of consensus purity for all the 7864 tumors should be presented.
2. It seems that LASSO was not the one with best overall performance in figure s3, the authors should clarify why LASSO regression model was chosen for modelling.
3. From figure 2c, the authors should explain why 117 samples were selected for each cancer type in the cross validation process and provide details of "cross-validated with cancer type as folds".
4. In figure 5b, it seems that tumors from independent datasets showed a relative high purity of larger than 0.25. The authors should evaluate their PUREE on different range of purity, especially the performance on low purity tumors. In addition, benchmarking of different methods would benefit from encompassing cancer types out of the 20 TCGA solid cancer types used in model training.
5. Page 7: "Next, we evaluated the ability of PUREE to predict purity in cancer types ... from the training data". The authors only used TCGA data to evaluate accuracy and generalization of PUREE, they should provide further estimation of robustness and accuracy of PUREE using three external validation datasets of lung and colorectal cancers.

6. The authors should provide detailed description of normalization method.

General response to reviewers

We thank the reviewers for the supportive feedback and valuable suggestions, which have significantly improved the manuscript. Based on this feedback, we have added several new analyses to the revised manuscript. In addition to addressing the reviewers' questions, we have improved the gene expression data pre-processing (gene filtering and data normalization), resulting in improved performance across independent test cohorts. These updates are described in the section 'Updates to method'. The general results and conclusions of our work remain unchanged, demonstrating that PUREE outperforms existing approaches for tumor purity estimation on both TCGA and a range of independent test cohorts. Here are some of the major highlights of the revised manuscript and response to reviewers' comments:

- An addition of **4 new datasets and 2 new tumor types** (paraganglioma and testicular cancer), each containing >100 samples, to the benchmark and external validation (requested by reviewer 3).
- An improvement to the data pre-preprocessing component of PUREE, resulting in **a 3% increase** in performance (median Pearson correlation and RMSE) on the external cohorts.
- We have included an **additional method** (DeconRNASeq) to the benchmark (requested by reviewer 1).
- An inclusion of **a new analysis** (supplementary figure S6), demonstrating that PUREE generally outperforms cancer-specific models, while also highlighting the degree of improvement for various cancer types (requested by reviewer 2)
- An inclusion of **a new analysis** (supplementary figure S9), showing that PUREE robustly predicts purity across tumors with potentially dissimilar stroma (requested by reviewer 2).
- An inclusion of **a new analysis** (supplementary figure S10), showing that PUREE outperforms existing methods for both low and high purity tumors (requested by reviewer 3).
- PUREE is now available as a **web service** (<https://puree.genome.sg/>) and a respective **Python API** (<https://github.com/skandlab/PUREE>).

We have rewritten parts of the manuscript for clarity in response to the comments by reviewers, and we have highlighted changes in **blue** in the revised manuscript. Below we have included a point-by-point response to all reviewer questions.

Method changes and performance improvements	3
Reviewer 1 Comments	4
Reviewer 1, General Comments	4
Reviewer 1, Comment 1	4
Reviewer 1, Comment 2	4
Reviewer 2 Comments	7
Reviewer 2, General Comments	7
Reviewer 2, Comment 1	7
Reviewer 2, Comment 2	7
Reviewer 2, Comment 3	7
Reviewer 2, Comment 4	8
Reviewer 2, Comment 5	8
Reviewer 2, Comment 6	9
Reviewer 2, Comment 7	9
Reviewer 2, Comment 8	10
Reviewer 2, Comment 9	12
Reviewer 2, Comment 10	12
Reviewer 2, Comment 11	14
Reviewer 2, Comment 12	15
Reviewer 3 Comments	16
Reviewer 3, General Comments	16
Reviewer 3, Comment 1	16
Reviewer 3, Comment 2	17
Reviewer 3, Comment 3	18
Reviewer 3, Comment 4	19
Reviewer 3, Comment 5	19
Reviewer 3, Comment 6	21

Method changes and performance improvements

We have made improvements to the data preprocessing pipeline of PUREE since the original manuscript submission. These changes have consistently improved the method's performance across datasets, improving both the median correlation and median RMSE by 3% across the independent test sets (see figure below).

Performance of updated PUREE method. Performance (correlation [left] and RMSE [right]) of the original version of PUREE (first submission) as well as the updated version (revised manuscript) on 8 external test sets.

Briefly, the changes to the pre-processing pipeline are as follows: inclusion only of genes encoded on autosomal chromosomes (chr1-22), filtering of genes with negative expression values (observed in rare instances in some cohorts), and use of a simpler rank-percentile normalization scheme (see Methods in the revised manuscript). As a result of these changes, the input feature set now contains 9554 genes (10037 previously), and the reduced predictive feature set used in PUREE now contains 158 genes (170 previously); all related figures and manuscript text have been updated (marked in blue) to reflect these changes.

Other minor changes to manuscript (not requested by reviewers)

- Previously, the Gradient Boosting model and the NuSVR model were trained with one fixed set of hyperparameters. We have updated their training procedure to pick correct hyperparameters using 5-fold cross-validation for a more fair benchmark.
- Linseed's benchmark was improved (as the LinSeed method returns two different cellular proportions without specifying what are the cell types being deconvolved, we now select the one that results in the best correlation/RMSE with the DNA-based purities).
- The external test cohort (CRC-GIS) has been made available publicly (<https://www.nature.com/articles/s41588-022-01100-4>). It is renamed as "Joanito et al." and the appropriate citation is provided in the revised manuscript.
- The time-memory benchmark from Figure 5 has been moved to the supplementary as Supplementary Figure 8.
- All changes are marked in blue in the revised manuscript.

Reviewer 1 Comments

Reviewer 1, General Comments

Reviewer Comment	PUREE is a weakly supervised machine algorithm to infer tumor purity from a tumor gene expression profile. Using LASSO regression, the algorithm first selects the most relevant features (genes) to be used in a pan-cancer model able to predict tumor purity. Trained on a set of 8000 TCGA samples for which genomic estimates of tumor purity are available, PUREE was the best performer among 5 different deconvolution algorithms tested and also on scRNA-Seq data.
Author Response	We thank the reviewer for taking the time to study our method and for raising some important questions. Below we provide a detailed point-by-point response for all the comments.

Reviewer 1, Comment 1

Reviewer Comment	While the methodology used is appropriate, the impact and novelty of the method is limited since many other methods are already available for this task, both from an algorithmic and technical perspective. For example, why using expression data to infer tumor purity if genomic data gives you a more straight forward answer for the same question? Performing RNA-Seq is still a relatively expensive method that doesn't work well for samples conserved for example in parafine blocks.
Author Response	We agree with the reviewer that tumor purity inferred from genomic data (e.g. using such methods as ABSOLUTE) is a more widespread approach. However, when both genomic and transcriptomic data are available for tumor samples, an accurate transcriptomic-based method such as PUREE could provide an additional and orthogonal/independent tumor purity estimate (e.g. Aran et al., Nature Communications, 2015; Ghosdastider et al., Cancer Research, 2021). Additionally, DNA and RNA are often extracted from different aliquots of a tumor , further highlighting the value of RNA-based purity estimation in such settings. We agree with the reviewer that RNA-seq on FFPE samples is challenging, however, technologies and protocols are constantly improving. As an example, we have recently noticed several large clinical cohorts studying response to immunotherapy (e.g. Mariathasan et al., Nature 2018; Liu et al. Nature Medicine 2019; Gide et al., Cancer Cell 2019; and Jung et al., Nature Communications 2019), which have all generated RNA-seq data from FFPE samples. Finally, since commonly used cell-type deconvolution algorithms (such as Cibersort and EPIC) also need to co-estimate purity to infer cell-type fractions in a tumor sample, PUREE could potentially also be adopted in these settings for improved cell-type deconvolution .
Edits to the manuscript	We have further clarified these points in the “Discussion” section: “We envision PUREE to be especially useful in settings where the DNA-seq data is either hard to obtain or absent. Even when tumor DNA-seq data is available, PUREE may provide an additional and orthogonal approach to tumor purity estimation. This may be especially relevant in cohorts and settings where the DNA and RNA are extracted from different aliquots of a tumor.”

Reviewer Comment
On the other hand, the authors only compare to methods based on bulk RNA-Seq. In the last years, deconvolution methods based on scRNA-Seq have been shown to outperform the methods used in the paper for benchmarking. These scRNA-Seq deconvolution methods can estimate not only tumor purity but also the percentage of each cellular type present in a tumor sample, allowing for instance quantification of tumor infiltrated lymphocytes and their type, what is highly relevant for therapeutic handling of the patients. Some of these methods such as MUSIC, Bisque and DECONRNASeq are summarized in recent reviews and would need to be included in the comparison.

Author Response
 We thank the reviewer for raising this point. We developed PUREE as a tumor-agnostic method that can predict tumor purity directly from a bulk tumor gene expression profile, without the need for additional and auxiliary experiments and datasets. Our benchmark therefore focused on methods that only require a bulk tumor gene-expression profile as input. Two of the methods that the reviewer listed (Bisque and MUSIC) have an additional input requirement in the form of a scRNA-seq dataset obtained from the tissue/tumor-type of interest. Since we don't have access to such scRNA-seq datasets for the 20+ tumor types in our benchmark we could not run these two methods. However, similar to CIBERSORTx, DeconRNASeq only requires a cell type-expression reference matrix and thus we were able to run it with the same reference matrix we used for CIBERSORTx. In the TCGA test-set, DeconRNASeq had the worst correlation and second-worst RMSE of all methods (see figure below and updated Figure 4 in manuscript).

Figure 4: Comparing PUREE and existing methods on TCGA data. a,b,c) Comparison of PUREE and 6 existing transcriptome deconvolution methods. Mean Pearson's correlation (a) and RMSE (b) of methods per cancer type when compared with genomic tumor purity estimates on the

	TCGA test data split (1573 samples); (c) shows aggregated mean correlation and RMSE metrics per cancer type for each method.
Edits to the manuscript	We have updated the main Figures 4 and 5, and the Supplementary Figures S1 and S8-S11 to include the DeconRNASeq method. Additionally, the description of the DeconRNASeq has been added to the Introduction section of the text and the Supplementary part. “Similarly, DeconRNASeq solves a non-negative least squares problem using a pre-defined cell-type signature matrix to derive the cellular proportions (33).”

Reviewer 2 Comments

Reviewer 2, General Comments

Reviewer Comment	The manuscript by Revkov et al. entitled "Accurate pan-cancer tumor purity estimation from gene expression data" uses a weakly supervised learning approach to infer tumor purity using gene expression profiles. Their method (PUREE) was used in 8000 bulk samples and single-cell datasets.
Author Response	We thank the reviewer for raising many important questions. Please refer to the below for our detailed answers.

Reviewer 2, Comment 1

Reviewer Comment	Page 2, second paragraph: Why the pathologist variation? Can you provide an estimate of the intra- and between observer variability? if available.
Author Response	The observation that pathology estimates of tumor purity from H&E slides can be inaccurate comes from the referenced paper by Smits et al. 2013 (DOI: 10.1038/modpathol.2013.134). This study evaluated the variability and accuracy of estimates from nine pathologists. The study measured tumor purity using a categorical 11-step scale. Briefly, the authors reported a difference of more than 6 categories (~50% difference in tumor purity) between the lowest and highest purity estimates from different pathologists. We hope that interested readers might investigate this topic further by following the reference in our text.

Reviewer 2, Comment 2

Reviewer Comment	Page 2, second paragraph: you forgot to mention the use of DNA methylation methods such as InfiniumPurify, MethylCIBERSORT, etc.
Author Response	We thank the reviewer for providing these additional relevant sources.
Edits to the manuscript	We have added references to DNA methylation-based methods InfiniumPurify and MethylCIBERSORT to the indicated part in the Introduction section: "More recent computational approaches to estimate tumor purity are based on DNA sequencing data where variation in allele frequencies of somatic DNA mutations, copy-number alterations (CNAs), or DNA methylation patterns are used to infer the malignant cell proportion (5,12–18)."

Reviewer 2, Comment 3

Reviewer Comment	Page 2 end of page 4: high correlation and low RMSE vs. what standard? Here you mention the silver standard; is this the CPE? this needs more explanation here.
Author Response	We thank the reviewer for highlighting this paragraph and the need for further clarification. The silver standard used throughout the manuscript is the tumor purity consensus estimates obtained from four different genomics-based methods (ABSOLUTE, AbsCNSeq,

	ASCAT and PurBayes). The correlation and RMSE metrics are calculated using these orthogonal genomics-based consensus estimates, derived from the DNA sequencing data of the same tumor samples.
Edits to the manuscript	We have further clarified this aspect in the revised manuscript: “The resulting method, PUREE, is able to robustly predict purity values with high correlation and low root mean squared error (RMSE) when compared to consensus genomics-based estimates from the same samples, outperforming existing deconvolution methods both on a TCGA test set (0.2 increase in Pearson’s correlation and 0.17 decrease in RMSE compared to the respective second best approaches) and seven external validation datasets of lung, colorectal, uterine, paraganglioma, and testicular cancers.”

Reviewer 2, Comment 4

Reviewer Comment	Suppl Fig S1: If possible you should signal which methods are DNA-based vs RNA-based. Also please include the n for the training and testing.
Author Response	We appreciate the reviewer’s feedback on this figure.
Edits to the manuscript	We have added an indication of which methods are DNA-based (genomics-based) and which are RNA-based to Supplementary Figure 1. We have also indicated the sample sizes for the training and testing splits (6219 and 1573 respectively).

a Tumor purities on TCGA samples (train split)

b Tumor purities on TCGA samples (test split)

Suppl. Figure S1: Agreement of the genomics- and transcriptomics-based purity estimation methods. Pairwise correlations of tumor purity estimates were computed separately for **a)** TCGA train (6291 samples) and **b)** TCGA test (1573 samples) splits of the dataset to account for the machine learning training procedure of PUREE.

Reviewer 2, Comment 5

Reviewer Comment	Results: The actual number is 7864 tumors? You should use that in your abstract or explain why some samples were excluded. Similarly, use the actual number of transcripts; is it 10K or 10037? Please be concordant through the text.
Author	The actual number of tumors used in the TCGA dataset in 7864, which is based on the

Response	sample set described in the publication by Ghoshdastider et al. (https://doi.org/10.1158/0008-5472.CAN-20-2352 , cited in the manuscript). The actual number of transcripts used is 9554 (after the in-review update), which is also referred to as the “10K” feature set. We have proofread and updated the indicated mentions of samples and feature sizes in the text to be more consistent.
Edits to the manuscript	We have rewritten the relevant parts of the text to state the exact number of samples and either the exact number of genes or refer to the “10K” feature set. For example: “From the 60,000 transcripts profiled in TCGA, we further selected and focused on 9,554 (10K) highly expressed protein-coding autosomal genes for model development (Methods).”

Reviewer 2, Comment 6

Reviewer Comment	Pages 4 and 5: Could you check the order of your supplementary figures? Why does S2 appear in the text after S3?
Author Response	We are grateful to the reviewer for noticing the discrepancy.
Edits to the manuscript	We have updated the order of the supplementary figures for better clarity, including changing the order of figures S2 (now “Exploring the range of machine learning methods for the PUREE approach”) and S3 (now “Hallmark pathway analysis of 158 genes used in the final model”).

Reviewer 2, Comment 7

Reviewer Comment	Page 4, supply Fig. S3: you should include some numbers about the selection of Lasso as your final model. What is the magnitude of the difference vs other methods? what is the range of errors in your estimates?
Author Response	After introducing the improvements to the gene expression data pre-processing in the revised manuscript, we have selected Lasso as the final predictive model for the 10K feature set (9554 features) and Linear Regression as the final predictive model for the reduced feature set (158 features). For the reduced feature set, all of the linear models showed comparable performance on the test set. NuSVR had slightly better performance than the linear models on the TCGA test set (Figure S2c,d, included below). However, NuSVR had a significantly larger train-vs-test performance difference for both correlation and RMSE. For example, NuSVR showed a 0.065 drop in median correlation and 0.016 median increase in RMSE when moving from the training to test set, whereas Linear Regression only showed a drop of 0.038 in median correlation and 0.005 increase in RMSE. This indicated a potential lower degree of generalization for NuSVR when applied to unseen data. We have decided to be conservative in our model selection and pick Linear Regression as a simple model with high accuracy and low train-to-test performance difference. As a measure of the range of errors in the models’ estimates, we have additionally provided standard deviations next to each boxplot (Figure S2, see below). As a measure of the

magnitude of the difference between methods, we have indicated differences between medians of the linear regression and nuSVR models for each feature set.

Edits to the manuscript

Supplementary Figure S2 has been updated to include standard deviations next to each boxplot; the text under the figure is updated and plots showing the “train-test” difference metrics are added.

Suppl. Figure S2: Exploring the difference between machine learning methods for the PUREE approach. c,d models trained on the reduced feature set of 158 features (c - Pearson's correlation, d - RMSE); linear regression was chosen over NuSVR due to comparable performance, lower train-test difference in performance and simplicity. Each column depicts performance on the TCGA test set and the difference in performance between train and test set respectively. Gradient Boosting, NuSVR, Ridge, Lasso and Elastic Net models were trained using 5-fold cross-validation hyperparameter selection procedure. Numbers next to boxplots indicate standard deviation. Delta between boxplots medians and Wilcoxon signed-rank test (two-tailed) shown for two best models in each plot (NuSVR and Lasso for 10K and NuSVR and Linear Regression for the reduced set).

Reviewer 2, Comment 8

Reviewer Comment

Figure 2, page 5-6: The selection of the targets based on the extreme purities is very smart. However, the question is whether the lower purity samples are homogeneous across different cancer types? What happened in tumors with very dissimilar stroma (SKCM, PAAD, GBM, LGG, or UCEC) is this affecting the estimates?

Author Response

We have performed a new analysis focusing on this question for the 5 listed cancer types. We compared the accuracy of PUREE and the other methods for these 5 cancer types relative to the remaining 15 tumor types. Confirming our existing observations, PUREE outperformed the other transcriptomics-based methods both on the 5 cancer types with

potentially higher levels of dissimilar stroma (SKCM, PAAD, GBM, LGG, or UCEC) as well as the other 15 tumor types. Furthermore, we noted that the average performance of PUREE on the 5 cancer types with dissimilar stroma (median correlation 0.82, median RMSE 0.093) is very similar to the performance of PUREE on the other 15 cancer types (median correlation 0.78, median RMSE 0.095), which suggests that the method is robust to potential differences in stromal composition and tissue of origin.

Manuscript edits

Added Supplementary Figure 9 showing the performance of PUREE and the other transcriptomics-based methods for the selected cancer types.

Supplementary Figure S9: Performance of PUREE and 6 other transcriptomics-based methods on TCGA test set for selected cancer types. Mean Pearson's correlation and RMSE of methods per cancer type when compared with genomic tumor purity estimates on the TCGA test data split (1573 samples); a) data shown for cancer types with potentially high levels of dissimilar stroma (GBM, LGG, PRAD, SKCM, UCEC), b) data shown for other 15 cancer types.

Also added a sentence to the "Benchmarking methods on independent datasets" section:

"We additionally evaluated the performance of methods when tested across solid tumor types with likely distinct stromal composition (e.g. brain cancers and skin cancers). Consistent with our previous observations, PUREE outperformed other transcriptomics-based approaches, showing comparable high accuracy across cancer types with expected dissimilar stromal composition (Suppl. Fig. 9)."

Reviewer 2, Comment 9

Reviewer Comment	Page 7, and Fig S8: Although CPE is pretty commonly used as a silver standard, some authors consider that the ESTIMATE could be a better proxy. Could you briefly expand on why that is the case?
Author Response	As highlighted in our response to the reviewer's comment 3, the silver standard used throughout the manuscript is the tumor purity consensus estimates obtained from four different genomics-based methods (ABSOLUTE, AbsCNSeq, ASCAT and PurBayes). The correlation and RMSE metrics are calculated using these orthogonal genomics-based consensus estimates, derived from the DNA sequencing data of the same tumor samples. The main motivation for using these genomics-based consensus estimates is that they provide an orthogonal and independent purity estimate that we can use to train and benchmark transcriptomics-based purity prediction methods (such as ESTIMATE and PUREE). In our benchmark (Figure 4 and 5), we compare the performance of PUREE and ESTIMATE. A key result of our paper is that PUREE displays higher concordance with these genomics-based consensus purity estimates across cancer types and independent test cohorts.

Reviewer 2, Comment 10

Reviewer Comment	Figure 3: The colors are very hard to read, and probably they are not color blind friendly. I suggest that you use a different palette. In addition to that can you expand on why SKCM is better in cancer-specific models with higher correlation and lower RSME? Something similar happens with STAD, but it is hard to distinguish with the colors. Please explain why that is the case to the readers.									
Author Response	We thank the reviewer for raising these questions. We have updated the figure to improve readability. The pan-cancer model performs better than the cancer-type specific on STAD. The performance of the pan-cancer model on SKCM is indeed worse, but only slightly (0.02 lower correlation and 0.005 higher RMSE), see table below.     Corr.: pan-cancer / cancer-specific RMSE: pan-cancer / cancer-specific     SKCM 0.883 / 0.905 0.094 / 0.089   STAD 0.736 / 0.64 0.106 / 0.121    In general, we observe no systematic difference between the accuracy of the pan-cancer and cancer-specific models. For most (N=12/20) cancer types the pan-cancer model is better (e.g. STAD), but for some cancer types (N=6/20) the cancer-specific model is better (e.g. SKCM). To further address the reviewers comment, we have highlighted cancer types that have better performance with the cancer-type specific and pan-cancer models, respectively, in a new supplementary figure (Figure S6).		Corr.: pan-cancer / cancer-specific	RMSE: pan-cancer / cancer-specific	SKCM	0.883 / 0.905	0.094 / 0.089	STAD	0.736 / 0.64	0.106 / 0.121
	Corr.: pan-cancer / cancer-specific	RMSE: pan-cancer / cancer-specific								
SKCM	0.883 / 0.905	0.094 / 0.089								
STAD	0.736 / 0.64	0.106 / 0.121								
Edits to the manuscript	We have updated Figure 3 and Figure 4, as well as corresponding supplementary figures S7, S8, S9, to make the cancer type-points more distinguishable and colorblind-friendly. Figure 3 is attached below as an example.									

Figure 3: PUREE versus cancer type-specific architectures and unseen cancer types. a) Performance (correlation and RMSE) of PUREE as compared to 20 cancer-type specific models on the test split of the TCGA dataset; **b)** Performance (Pearson's correlation and RMSE) of PUREE when predicting on a cancer type included in training data versus when the same cancer type was excluded from the training data. Differences were evaluated with the Wilcoxon signed-rank test (two-tailed), and delta is the difference between medians of metrics across all cancer types. P-values from Wilcoxon signed-rank test (two-tailed) shown.

New figure S6 highlighting the differences in performance between pan-cancer and cancer-specific models.

Reviewer 2, Comment 11

Reviewer Comment	Figure 6: Could you expand on the differences in the densities between those genes with positive and negative correlations with malignant cells? Could you also add the actual numbers?
Author Response	The distributions of Z-scores of malignant and non-malignant cells have noticeably different medians for both melanoma single-cell data (0.2 vs. -0.2 for positively correlated genes, -0.23 vs. 0.09 for negatively correlated genes) and head and neck single-cell data (0.19 vs. -0.11 for positively correlated genes, -0.012 vs. -0.02 for negatively correlated genes). We have indicated the medians of each distribution in the updated Figure 6.
Edits to the manuscript	Figure 6 is updated with the medians of each distribution.

Reviewer 2, Comment 12

Reviewer Comment	Page 12: are there any essential differences between cancer types that you should mention in this context? What about the differences with DNA-based method estimates?
Author Response	We assume the reviewer is referring to the fact that we use 20 different solid cancer types in our TCGA-based dataset, all of which exhibit different median purities. We have further clarified this aspect in the Discussion and Conclusion.
Edits to the manuscript	Updated the Discussion and Conclusion paragraphs related to the range of tumor purities in the dataset: "The model is trained using weak supervision and consensus tumor purity labels obtained from tumor DNA sequencing data spanning 20 solid tumor types. These tumor types displayed variable tumor purity, ranging from 35% (median) in pancreatic cancer to 72% in ovarian cancer."

Reviewer 3 Comments

Reviewer 3, General Comments

Reviewer Comment	The authors present a manuscript on a machine learning approach to estimate purity from a bulk tumor gene expression profile. The tool is based on a lasso regression classifier over 170 features derived by a stringent feature selection strategy and trained over 7864 TCGA solid tumors. The manuscript is concise, generally clear, and the results appear convincing.
Author Response	We thank the reviewer for the supportive feedback. Please find our responses to the comments below.

Reviewer 3, Comment 1

Reviewer Comment	The authors should clarify why only 7864 TCGA tumors were used and how they selected these tumors. Also, the distribution of consensus purity for all the 7864 tumors should be presented.		
Author Response	Our study is based on the dataset generated by Ghoshdastider et al. (https://doi.org/10.1158/0008-5472.CAN-20-2352). This study computed consensus genomic purity estimates across TCGA samples in these 20 cancer types, which we have here used to develop PUREE. The distribution of consensus purity is graphically presented in Figure 2a. To further address the reviewers comment, we have additionally added a detailed description of tumor purity distribution per cancer type to the supplementary.		
Edits to the manuscript	We have added 2 new columns called “number of samples” and “median tumor purity” to the description of the TCGA dataset in Supplementary Table 1 (presented below).		
	Abbreviation	Full name	Number of samples (in train set / in test set)
	BRCA	Breast invasive carcinoma	1060 (848 / 212)
	LUAD	Lung adenocarcinoma	508 (406 / 102)
	LGG	Brain lower grade glioma	506 (405 / 101)
	HNSC	Head and neck squamous cell carcinoma	494 (395 / 99)
	PRAD	Prostate adenocarcinoma	489 (391 / 98)
	THCA	Thyroid carcinoma	485 (388 / 97)
	LUSC	Lung squamous cell carcinoma	482 (385 / 97)
	STAD	Stomach adenocarcinoma	410 (328 / 82)
	BLCA	Bladder urothelial carcinoma	403 (322 / 81)
	KIRC	Kidney renal clear cell carcinoma	377 (302 / 75)
	CRC	Combined: colon adenocarcinoma (COAD) and rectum adenocarcinoma (READ)	367 (294 / 73)
	SKCM	Skin cutaneous melanoma	365 (292 / 73)
LIHC	Liver hepatocellular carcinoma	360 (288 / 72)	
		Median tumor purity (in train set / in test set)	
		0.56 (0.56 / 0.55)	
		0.42 (0.42 / 0.44)	
		0.66 (0.67 / 0.62)	
		0.47 (0.47 / 0.47)	
		0.48 (0.48 / 0.5)	
		0.52 (0.52 / 0.54)	
		0.49 (0.49 / 0.48)	
		0.47 (0.46 / 0.47)	
		0.56 (0.57 / 0.56)	
		0.51 (0.51 / 0.53)	
		0.62 (0.62 / 0.62)	
		0.64 (0.66 / 0.57)	
		0.66 (0.66 / 0.67)	

OV	Ovarian serous cystadenocarcinoma	301 (241 / 60)	0.72 (0.72 / 0.76)
CESC	Cervical squamous cell carcinoma and endocervical adenocarcinoma	292 (234 / 58)	0.61 (0.6 / 0.63)
KIRP	Kidney renal papillary cell carcinoma	284 (227 / 57)	0.66 (0.66 / 0.61)
ESCA	Esophageal carcinoma	180 (144 / 36)	0.56 (0.54 / 0.59)
UCEC	Uterine corpus endometrial carcinoma	179 (143 / 36)	0.67 (0.66 / 0.67)
PAAD	Pancreatic adenocarcinoma	176 (141 / 35)	0.35 (0.36 / 0.35)
GBM	Glioblastoma multiforme	146 (117 / 29)	0.72 (0.73 / 0.72)
Suppl. Table S1: TCGA dataset description and train/test set distribution. TCGA dataset used in this study was compiled from datasets of 20 solid cancer types, resulting in 7864 samples (6291 in the train set, 1573 in the test set). Median purity is rounded to 2 digits after the decimal point.			

Reviewer 3, Comment 2

Reviewer Comment	It seems that LASSO was not the one with best overall performance in figure s3, the authors should clarify why LASSO regression model was chosen for modelling.
Author Response	After introducing the improvements to the gene expression data pre-processing in the revised manuscript, we have selected Lasso as the predictive model for the 10K feature set (9554 features) and Linear Regression as the final predictive model for the reduced feature set (PUREE, 158 features). For the reduced feature set, all of the linear models showed comparable performance on the test set. NuSVR had slightly better performance than the linear models on the TCGA test set (Figure S2c,d, included below). However, NuSVR had a significantly larger train-vs-test performance difference for both correlation and RMSE. For example, NuSVR showed a 0.065 drop in median correlation and 0.016 median increase in RMSE when moving from the training to test set, whereas Linear Regression only showed a drop of 0.038 in median correlation and 0.005 increase in RMSE. This indicated a potential lower degree of generalization for NuSVR when applied to unseen data. We have decided to be conservative in our model selection and pick Linear Regression as a simple model with high accuracy and low train-to-test performance difference. As a measure of the range of errors in the models' estimates, we have additionally provided standard deviations next to each boxplot (Figure S2, see below). As a measure of the magnitude of the difference between methods, we have indicated differences between medians of the linear regression and nuSVR models for each feature set.
Edits to the manuscript	Supplementary Figure S2 has been updated to include standard deviations next to each boxplot; the text under the figure is updated and plots showing the "train-test" difference metrics are added.

Suppl. Figure S2: Exploring the range of machine learning methods for the PUREE approach. c,d) models trained on the reduced feature set of 158 features (c - Pearson's correlation, d - RMSE); linear regression was chosen over NuSVR due to comparable performance, lower train-test difference in performance and simplicity. Each column depicts performance on the TCGA test set and the difference in performance between train and test set respectively. Gradient Boosting, NuSVR, Ridge, Lasso and Elastic Net models were trained using 5-fold cross-validation hyperparameter selection procedure. Numbers next to boxplots indicate standard deviation. Delta between boxplots medians and Wilcoxon signed-rank test (two-tailed) shown for two best models in each plot (NuSVR and Lasso for 10K and NuSVR and Linear Regression for the reduced set).

Reviewer 3, Comment 3

Reviewer Comment	From figure 2c, the authors should explain why 117 samples were selected for each cancer type in the cross validation process and provide details of "cross-validated with cancer type as folds".
Author Response	To ensure all cancer types were represented with the same number of samples, we set the number of samples to the cancer type with lowest sample count (GBM, N=117) in the TCGA train set. In the second step of our feature selection strategy, we utilize a Lasso model trained on the full range of purities and cross-validated on cancer types as folds. Briefly, this is done by training on samples from N-1 cancer types, and testing on samples from the remaining withheld cancer type (leave-one-cancer-type out). We have clarified this point further in the revised manuscript.
Edits to the manuscript	Updated the text in the Methods part of the manuscript ("Feature selection"):

	“Next, we further iteratively selected features equally relevant to all cancer types by training a Lasso model (cross-validated with cancer type as folds) on a balanced subset of the full TCGA training set comprising all purity ranges, using the earlier selected 167 genes as initial features. Briefly, this is done by training on samples from N-1 cancer types, and testing on samples from the remaining withheld cancer type (leave-one-cancer-type out). We balance this step of feature selection by selecting only 117 samples from each cancer type per fold, as it is the least number of samples per cancer type (GBM, N=117) found in the TCGA train split.”
--	--

Reviewer 3, Comment 4

Reviewer Comment	In figure 5b, it seems that tumors from independent datasets showed a relative high purity of larger than 0.25. The authors should evaluate their PUREE on different range of purity, especially the performance on low purity tumors. In addition, benchmarking of different methods would benefit from encompassing cancer types out of the 20 TCGA solid cancer types used in model training.
Author Response	We appreciate the reviewer’s comment. We note that the external lung cancer cohort, Chen et al., used in the benchmark has a median tumor purity of 0.45, which provides an estimate of PUREE’s performance for samples with relatively low tumor purities. In this cohort, PUREE outperformed existing methods when considering both performance metrics (PUREE Pearson’s $r=0.62$ and $RMSE=0.14$; next-best methods were ESTIMATE [$r=0.59$, $RMSE=0.23$] and EPIC [$r=0.54$; $RMSE=0.12$]). To further address the reviewer's comment, we analyzed the performance in the 5 TCGA cancer types with the lowest and highest median consensus purities, respectively. PUREE consistently outperformed the other methods, and demonstrated a significant improvement over the next best approach in the group of cancer types with the lowest median purity (PUREE [$r=0.79$; $RMSE=0.09$], CIBERSORTx [$r=0.58$; $RMSE=0.18$]).

Supplementary Figure S10: Performance of PUREE and 6 other transcriptomics-based methods on TCGA test set for selected cancer types with different median purities. Mean Pearson's correlation and RMSE of methods per cancer type when compared with genomic tumor purity estimates on the TCGA test data split (1573 samples); a) cancer types with the lowest median consensus tumor purity (HNSC, LUAD, PAAD, PRAD, STAD), b) cancer types with the highest median consensus tumor purity (GBM, LGG, LIHC, OV, UCEC).

To address the last point, we have extended our benchmark with 4 additional test cohorts and 2 new cancer types (paraganglioma and testicular cancers). Please see our response to the next comment (Comment 5) for details.

Edits to the manuscript

Added a new Supplementary Figure S10.

Added a sentence to the Benchmarking of methods on independent datasets section: "A similar analysis showed that PUREE outperformed the other methods on the cancer types with extreme median tumor purities (Suppl. Fig. S10)."

Added a new Supplementary Table S2 with the brief descriptions of the independent test cohorts and the median purities indicated.

Cohort name	Cancer type	Num. of samples	Median purity	Purity estimation method
TCGA (test set)	20 solid cancer types	1573	0.56	Consensus (ABSOLUTE, ASCAT, PurBayes,

				AbsCNSeq)
Chen et al.	Lung	172	0.45	Sequenza
Chua et al.	Lung	64	0.6	Consensus (theta2, TitanCNA, PurBayes, AbsCNSeq)
Joanito et al.	Colorectal	153	0.62	Consensus (theta2, TitanCNA, PurBayes, AbsCNSeq)
TCGA-CRC+	Colorectal	243	0.7	ABSOLUTE
TCGA-UCEC+	Uterine	353	0.76	ABSOLUTE
TCGA-PCPG	Paraganglioma	164	0.74	ABSOLUTE
TCGA-TGCT	Testicular	155	0.6	ABSOLUTE

Suppl. Table S2: Independent datasets descriptions and purity statistics. Median purity is rounded to 2 digits after the decimal point.

Reviewer 3, Comment 5

Reviewer Comment	Page 7: "Next, we evaluated the ability of PUREE to predict purity in cancer types ... from the training data". The authors only used TCGA data to evaluate accuracy and generalization of PUREE, they should provide further estimation of robustness and accuracy of PUREE using three external validation datasets of lung and colorectal cancers.
Author Response	In the original submission, we benchmarked PUREE and 5 other methods on 3 non-TCGA datasets of colorectal (Joanito et al., N=153 samples) and lung cancers (Chen et al., N=172, and Chua et al., N=64). In agreement with the TCGA test set results, PUREE outperformed existing methods across these external datasets (Figure 5, Supplementary Figure S11). To further address the reviewers' comments, we have added 4 new datasets and 2 new cancer types which were not present in the TCGA data used for training and testing. These are TCGA-CRC+ (additional TCGA colorectal cancer samples, N=243 samples), TCGA-UCEC+ (additional uterine endometrial cancer samples, N=353), TCGA-TGCT (testicular cancer, N=155), TCGA-PCPG (paraganglioma, N=164). For these cohorts we used DNA-based purity estimates computed by the ABSOLUTE algorithm (provided by the TCGA consortium) as pseudo-ground truth estimates. Briefly, PUREE outperformed all other methods on colorectal, uterine and paraganglioma cancer cohorts (Pearson's $r \geq 0.7$, RMSE ≤ 0.12). On the testicular cancer cohort PUREE demonstrated favorable performance when considering both performance metrics (PUREE [Pearson's r 0.62; RMSE 0.18], ESTIMATE (next-best method) [Pearson's r 0.69; RMSE 0.24]).
Edits to the manuscript	Main figure 5 is edited, now using 4 additional external cohorts. a) and b) now show mean RMSE and Correlation per external cohort. Time-memory benchmark is moved to Supplementary Figure S12 for clarity.

Figure 5: Benchmarking of methods on independent datasets. Comparing the purity estimates of PUREE and 6 existing transcriptome deconvolution methods. The predictions from each method were compared with genomic tumor purity estimates across 7 cohorts: the TCGA test set, lung cancer samples from Chen et. al (35), lung cancer samples from Chua et. al (36), a colorectal cancer cohort from Joanito et al. (37), and 4 TCGA cohorts of colorectal (CRC+), uterine endometrial (UCEC+), pheochromocytoma and paraganglioma (PCPG), and testicular cancer (TGCT) not present in the TCGA dataset used for training and testing. **a, b**) Mean Pearson's correlation (a) and RMSE (b) per cohort for all methods. **c**) Mean correlation and RMSE per cohort of every method shown together. DeMixT could not be run on the Chua et al., Joanito et al., PCPG and TGCT cohorts due to the absence of normal samples in these cohorts.

Figure S11 has been updated with 4 new datasets, showing Pearson's r and RMSE of different transcriptomics-based methods on the 8 test datasets (TCGA-test split and 7 independent cohorts). Additionally, all the relevant text in the main manuscript and the supplementary have been updated.

Reviewer 3, Comment 6

Reviewer Comment	The authors should provide detailed description of normalization method.
Author Response	We thank the reviewer for this comment. We have improved the description of the normalization strategy in the Methods of the revised manuscript.
Edits to the	We have provided a detailed description of the normalization strategy in the Method section ("Gene expression data rank-transformation"):

manuscript	“Gene expression data is rank-percentile normalized (sample-wise) when serving as input for PUREE. The initial rank-transformation allows for generalization across different gene expression platforms (e.g. RNA-seq, microarray) and measurement units (e.g. Transcripts Per Million (TPM), Fragments Per Kilobase Million (FPKM)). Briefly, gene expression values (e.g. TPMs [0, 0, 1, 5, 100]) are first ranked based on their position within a sample in ascending order, assigning the lowest possible rank for tied groups ([1, 1, 3, 4, 5]). The percentile of the resulting rank is then computed ([0.2, 0.2, 0.6, 0.8, 1]). The resulting percentiles computed relative to the ranking universe of the 10K feature set serve as input values to PUREE.”
------------	--

REVIEWERS' COMMENTS:

Reviewer #1 (Remarks to the Author):

The manuscript has been substantially improved and the authors have satisfactorily answered to my questions.

Reviewer #2 (Remarks to the Author):

The manuscript by Revkov et al. entitled "Accurate pan-cancer tumor purity estimation from gene expression data" uses a weakly supervised learning approach to infer tumor purity using bulk gene expression profiles. They have carefully added explanations to the reviewers concerns and have now clarified several aspects int heir manuscript.

I have no additional comments.

Reviewer #3 (Remarks to the Author):

I appreciate the authors' detailed point-by-point response, which mostly answered my questions. A small question to response to my comment 3, how do you select the 117 samples from each caner type, randomly or with certain criteria?

Response to reviewers (Round 2)

Reviewer 1 and Reviewer 2

Reviewers Comments	Reviewer 1: "The manuscript has been substantially improved and the authors have satisfactorily answered to my questions." Reviewer 2: "The manuscript by Revkov et al. entitled "Accurate pan-cancer tumor purity estimation from gene expression data" uses a weakly supervised learning approach to infer tumor purity using bulk gene expression profiles. They have carefully added explanations to the reviewers concerns and have now clarified several aspects int heir manuscript. I have no additional comments."
Author Response	We thank the reviewers for the favorable feedback, and we were grateful for the provided opportunity to improve our manuscript.

Reviewer 3

Reviewer Comment	I appreciate the authors' detailed point-by-point response, which mostly answered my questions. A small question to response to my comment 3 , how do you select the 117 samples from each caner type , randomly or with certain criteria?
Author Response	We are glad that we were able to answer most of the reviewers' questions. Regarding the last point that the reviewer is raising, the sample selection for the balanced cross-validation was made in a way to preserve the original tumor purity distribution of each cancer type based on the training split. More specifically, for each cancer type the respective tumor purity range was binned into 10 equal intervals, and the proportions of samples in each interval were used as probabilities for choosing the resulting 117 samples.
Edits to the manuscript	We have edited the sentence in the "Feature selection" section, clarifying how the samples were selected for the balanced cross-validation: "We perform balanced feature selection by selecting 117 samples (determined by tumor type with the lowest number of samples in the training set: GBM, N=117) from each cancer type while preserving the original purity distribution"